# Myrtaceae in Australia: Use of Cryobiotechnologies for the Conservation of a Significant Plant Family under Threat

**DOI:** 10.3390/plants11081017

**Published:** 2022-04-08

**Authors:** Lyndle K. Hardstaff, Karen D. Sommerville, Bryn Funnekotter, Eric Bunn, Catherine A. Offord, Ricardo L. Mancera

**Affiliations:** 1Curtin Medical School, Curtin University, GPO Box U1987, Perth, WA 6845, Australia; b.funnekotter@curtin.edu.au (B.F.); eric.bunn@dbca.wa.gov.au (E.B.); 2The Australian PlantBank, Australian Institute of Botanical Science, The Royal Botanic Gardens and Domain Trust, Mount Annan, NSW 2567, Australia; karen.sommerville@botanicgardens.nsw.gov.au (K.D.S.); cathy.offord@botanicgardens.nsw.gov.au (C.A.O.); 3Kings Park Science, Department of Biodiversity, Conservation and Attractions, Kings Park, WA 6005, Australia

**Keywords:** ex situ conservation, cryobiotechnology, cryostorage, plant tissue culture, in vitro culture, exceptional species, *Austropuccinia psidii*

## Abstract

The Myrtaceae is a very large and diverse family containing a number of economically and ecologically valuable species. In Australia, the family contains approximately 1700 species from 70 genera and is structurally and floristically dominant in many diverse ecosystems. In addition to threats from habitat fragmentation and increasing rates of natural disasters, infection by myrtle rust caused by *Austropuccinia psidii* is of significant concern to Australian Myrtaceae species. Repeated infections of new growth have caused host death and suppressed host populations by preventing seed set. Although most Myrtaceae species demonstrate orthodox seed storage behavior, exceptional species such as those with desiccation sensitive seed or from myrtle rust-suppressed populations require alternate conservation strategies such as those offered by cryobiotechnology. Targeting seven key Australian genera, we reviewed the available literature for examples of cryobiotechnology utilized for conservation of Myrtaceae. While there were only limited examples of successful cryopreservation for a few genera in this family, successful cryopreservation of both shoot tips and embryonic axes suggest that cryobiotechnology provides a viable alternative for the conservation of exceptional species and a potential safe storage method for the many Myrtaceae species under threat from *A. psidii*.

## 1. Introduction

The Myrtaceae is a large and diverse family found on all continents except Antarctica, though occurring mainly in tropical and temperate regions of the Southern Hemisphere [1]. The family includes more than 6000 species [2] and new species continue to be discovered in remote tropical forests, with many still to be formally identified to species level [3]. Many species have been cultivated for their timber, oils, or fruits and are economically valuable both within their country of origin and in plantations around the world. On-going impacts of land clearing, a changing climate, and myrtle rust on this family mean there is a growing need to conserve species and cultivars ex situ. While most dryland genera can be conserved effectively by seed banking, species that no longer produce seeds, or that have desiccation- or freezing-sensitive seeds, require alternative conservation techniques such as in vitro culture and cryopreservation. In this review we discuss the significance of the Myrtaceae family (globally and in Australia), on-going threats to species in the wild and in cultivation, and possible ex situ conservation measures to mitigate extinction risk. We then review the available literature for information on cryobiotechnology tools that have been utilized to preserve exceptional species in the Myrtaceae, and outline a research approach for developing such tools for highly threatened Australian species.

## 2. Significance of Myrtaceae

Given the size and diversity of the Myrtaceae family, it is not surprising that there is a similar diversity of ways in which species have been exploited for human use (Table 1). Products from some species have been utilized for thousands of years by indigenous populations for food, shelter, bedding, transport, medicines, toys and tools [4,5,6,7]. Potentially useful properties of a range of species continue to be investigated, particularly phytochemicals for use in medicine and food e.g., reviews by [8,9].

*Eucalyptus* species and their cultivated crosses are perhaps the most widely planted of the Myrtaceae, with both small-scale and industrial plantations covering millions of hectares globally, particularly in Asia and South America [1,10]. Different *Eucalyptus* species can be grown in a range of geographical and climatic conditions and are suitable for the production of fuel, timber, pulp, tannin, oils, windbreaks, ornamental plants and honey [36]. Some of the most valuable essential oils are also products of Myrtaceae, including clove (*Syzygium aromaticum*), tea tree (*Melaleuca alternifolia*), lemon-scented (*Backhousia citriodora* and *Corymbia citriodora*), and eucalyptus peppermint (various *Eucalyptus* spp.) oils [1]. These oils are used in everything from food production and cosmetics to medicines and pesticides. Species such as guava (*Psidium guajava*), rose apple (*S. jambos*), wax apple (*S. samargarense*) and jamun (*S. cumini*) are important horticultural species cultivated in tropical regions for their fruit [37,38,39].

In Australia, the Myrtaceae family is structurally and floristically dominant in many ecosystems [40], occurring in 11 of the 13 major plant formations recognized by Specht [41]. Approximately 1700 species from 70 genera of Myrtaceae are found growing in diverse habitats [42], from rainforests to arid regions and lowland swamps to alpine regions [1]. Myrtaceae species contribute a high proportion to the plant biomass and diversity of the continent [41], and provide shelter, breeding sites, and food sources for a wide range of insectivorous and vertebrate fauna [10,41,43,44,45]. Drier forests and woodlands are dominated by eucalypts (species from the genera *Eucalyptus*, *Corymbia*, *Angophora* and *Syncarpia*) [10,43] and species from the Myrteae and Syzygieae tribes are common in tropical and subtropical forests [46]. Eucalypts are particularly important to a range of threatened fauna, providing the main food source for the koala (*Phascolarctos cinereus*) [47], an important component of the diet of the Eastern pygmy-possum (*Cercatetus nanus*) [48], and preferred nesting habitat for glossy black cockatoos (*Calyptorhynchus lathami*) and the superb parrot (*Polytelis swainsonii*) [49].

Myrtaceae species are also important to Australia’s small but growing ‘bush food’ industry. *Backhousia citriodora* (lemon myrtle), *Kunzea pomifera* (muntries), *Syzygium anisatum* (aniseed myrtle), and *S. leuhmannii* (riberry) are four of the most commercialized Myrtaceae, grown mainly in mixed-species plantations on the east coast of Australia [29]. A number of Australian fruits are gaining popularity as ‘functional’ foods due to their relatively low sugar contents and high nutrition levels, with both *K. pomifera* and *S. leuhmannii* listed as priority fruits for development [9] and further research of plant characteristics and production requirements [50]. The fruits of other species utilized by indigenous communities–such as *S. suborbiculare*, *S. eucalyptoides* and *Eugenia reinwardtiana* [7]–may also have value as bush foods. While large-scale production of Australian bush foods mostly occurs overseas, small-scale operations are valued for supporting indigenous communities, providing supplementary income in remote areas [15,29].

## 3. Threats to Myrtaceae—A Focus on Myrtle Rust

Though a number of Myrtaceae species are considered valuable in diverse industries around the world, many are also threatened in their natural habitats. For example, a review of plant conservation in Brazil [30] identified 14 Myrtaceae species from the Atlantic Forest biome as being in danger of extinction. This included 11 species with recalcitrant seeds, eight of them *Eugenia* spp., with value to their local ecosystems and for fruit production and the pharmaceutical industry [30].

Threats to Australian flora, including Myrtaceae species, include changing land use, habitat fragmentation, increasing rates of natural disasters–such as extreme fires and flooding–due to climate change, and invasive species [51,52]. Invasive fungal pathogens in particular have been implicated in the decline of a number of species globally, both as individual species and in multi-species interactions [53,54,55,56]. One such species causing concern in Australia and globally is *Austropuccinia psidii*, a fungal pathogen native to South America that causes a disease known as myrtle rust.

Although *A. psidii* typically causes only mild infections in species within its natural distribution [57,58], the impact of the pathogen on ‘naïve’ hosts in new environments has varied with pathogen strain and susceptibility of host species [53,59]. For example, rapid and severe damage was reported in the Jamaican allspice industry in 1934 within two years of myrtle rust detection [57]; whereas only two of 70 susceptible species in New Caledonia were severely affected in the first three years after detection of another strain [55]. Globally, over 500 species from 69 genera are known to have some susceptibility to *A. psidii* [60]. Many of these records are from areas with more recent incursions, for example New Caledonia in 2013 [55], South Africa in 2013 [61], Indonesia in 2015 [62], Singapore in 2016 [63], and New Zealand in 2017 [64]. Myrtle rust was first reported in Australia in 2010 as *Uredo rangelii* [65] and has since spread along the entire east coast with reported damage to both natural ecosystems and to industry [53,66]. Makinson [53] lists 394 host species in Australia, including 358 native species and subspecies, ranging from ‘relatively tolerant’ to having ‘extreme susceptibility’ to *A. psidii*. This list is likely to grow with predicted climatic changes increasing the potential distribution of the pathogen [67].

While only one strain of *A. psidii* is presently known in Australia, it has been shown to cause host death after repeated infections of new growth [53,58,66]. Myrtle rust may also suppress host populations by preventing seed set, and suppressing regeneration through infection of seedlings and suckers [68]. Less susceptible species may also succumb to the compounded effects of insect damage and *A. psidii*, for example as seen in *Melaleuca quinquenervia* and cultivars of *Syzygium* [68]. In the twelve years since *A. psidii* was detected in Australia, four species have been declared critically endangered as a direct result of myrtle rust [53,69] and, along with 12 other rainforest species, are at risk of extinction within one generation [70].

## 4. Ex Situ Conservation Efforts

Given on-going threats to Myrtaceae species in the wild, ex situ conservation is needed to preserve species from extinction and to provide a source of material for restoration. Ex situ conservation includes the maintenance of plant genetic diversity in living collections, seed banks, in vitro collections, and cryostorage. Each method has its own benefits and challenges, and these are outlined with many examples from Australia in Martyn Yenson, et al. [71]. Approximately 95% of the Myrtaceae are thought to have orthodox seed suitable for storage in a seedbank [72] and seedbanks in Australia currently hold collections for a total of 1534 taxa from 72 genera [73]. The remainder of the family falls into the category of ‘exceptional’ [74], with species that no longer produce seeds due to myrtle rust (e.g., *Rhodamnia rubescens* and *Rhodomyrtus psidioides* [70,75]), or that produce seeds that are intolerant of desiccation (e.g., *Syzygium* spp. [72,76]) or storage at −20 °C (e.g., *Backhousia citriodora* [77] and *Rhodamnia maideniana* [78]).

Woody plant species from rainforests are over-represented in the exceptional species categories, with a much higher proportion of species with desiccation-sensitive seeds than those from drier vegetation types [76,79,80]. Although the majority of Myrtaceae species have dry fruit and orthodox seeds, fleshy fruited trees and shrubs are common in the Myrteae and Syzygieae tribes [1] and these are more likely to have desiccation sensitive seeds than those with dry fruits [11]. A study of Australian species has found this to be true for fleshy-fruited species containing a single seed, however those containing a number of small seeds were more likely to show intermediate behaviors, with typically desiccation tolerant seeds but a range of responses to freezing [78]. As repeated infections of reproductive organs by myrtle rust are likely to affect seed set [81], even species with orthodox seed storage behavior may become exceptional species, resulting in the need for alternate conservation methods such as tissue culture and cryopreservation for a greater proportion of species.

## 5. Cryobiotechnology Applied to Myrtacaeae

Cryobiotechnology includes cryopreservation (storage of germplasm at ultra-low temperatures) and the in vitro technologies needed to support the preparation and recovery of cryopreserved tissues [82,83]. The storage of germplasm at ultra-low temperatures in liquid nitrogen limits any biochemical activity, extending the viability of the stored germplasm far longer than traditional seed banking as long as the temperatures are kept constant [84]. Species from multiple Australian plant families have been cryopreserved successfully [85]. Cryobiotechnology could thus provide a viable alternative for the conservation of exceptional Myrtaceae species, including the many species under threat from *A. psidii*. The remainder of this review will therefore focus on reports of cryobiotechnologies applied to seven key genera with known susceptibility to myrtle rust both in the wild and in cultivation, comprising: two very large genera, i.e., *Eucalyptus* and *Syzygium*; one small genus with species of commercial and ecological value, i.e., *Backhousia* [70]; and four genera with a number of species at risk of imminent extinction from myrtle rust, i.e., *Gossia*, *Lenwebbia*, *Rhodamnia*, and *Rhodomyrtus* [53,68,70].

### 5.1. Eucalyptus

With over 700 species and a number of hybrids and cultivars, *Eucalyptus* is one of the largest genera in the Myrtaceae [86]. Perhaps as a result of their typically orthodox seed, research into cryobiotechnology of this genus has been limited to conservation of valuable timber cultivars, hybrids, and elite clonal lines rather than ‘wild’ species [87]. In fact, ‘unknown genetic diversity’ in stored seed was listed as a problem for some authors aiming to conserve specific genotypes, limiting the applicability of these collections for conservation [88]. Historically, conservation of genetic resources in industry has utilized seeds, cuttings, and clonal hedges [89], however a number of research groups have begun to turn to biotechnology in recent decades.

In vitro cultures of *Eucalyptus* species have been comparatively well studied, with work done on optimizing establishment of shoot, callus and somatic embryogenic cultures, looking at the effects of basal salts and plant growth regulators required [90], with over 82 species and 19 hybrids mentioned in various publications [90,91,92]. In stark contrast, only 4 threatened species have been cultured in vitro [93,94,95]. A number of difficulties have been reported with in vitro initiation and maintenance of *Eucalyptus* cultures (from both seed and vegetative material), including browning and rapid dieback, internal and external contamination, low germination rates, slow growth, and limited multiplication [96,97,98]. Limited success with hardening off for planting out has also been noted as a barrier to mass production of *Eucalyptus* material from in vitro cultures [36].

The use of cryobiotechnology to preserve these in vitro collections has seen less application, with successful cryostorage limited to a few species and hybrids (Table 2). Although reported survival rates vary from 0–100%, the diversity of species and methods that have been trialed is promising (Table 2). Some reports have shown identical or even increased viability after storage in liquid nitrogen [99,100]. Others have shown high rates of survival after pretreatment prior to cryostorage (e.g., desiccation and application of cryoprotectants), but no survival after storage in liquid nitrogen [97,101]. Reduced survival after exposure to liquid nitrogen may be a result of high water content of stored material. For example, studies of *E. grandis* axillary buds have found the buds to be desiccation-sensitive. Although the use of ABA and encapsulation was seen to increase survival of buds after pretreatment, the authors cautioned that the material may not then be sufficiently desiccated for successful cryopreservation [89,102].

### 5.2. Syzygium

Distributed throughout tropical and subtropical regions around the globe, *Syzygium* contains at least 1119 species [86], of which approximately 70 are found in Australia. *Syzygium* species typically have fleshy fruits with desiccation sensitive seeds [72] however there is some variation within the genus. For example, *S. paniculatum* and *S. unipunctatum* have desiccation sensitive seeds [78] while *S. anisatum* fruits are dry and their seeds are orthodox, though artificial aging experiments have shown they are likely to be very short-lived in storage at −20 °C [106]. *S. anisatum* is the only species of the genus with seeds held in storage at the Australian PlantBank [73].

*Syzygium* species are also known to have a range of susceptibility to myrtle rust, ranging from ‘low’ to ‘very high’ [81]. *S. maire*, the only native species of this genus in New Zealand, is also known to be susceptible to myrtle rust and has been made a priority for conservation using cryobiotechnology [107,108]. With the exception of *S. maire*, much of the literature reporting use of cryobiotechnology for conservation of *Syzygium* species is focused on species cultivated for fruit and medicinal purposes, particularly those species common in Asia (e.g., *S. cumini* [109]). In Australia, in vitro collections of *S. francissi* have been successfully initiated from cuttings [110] and *S. paniculatum* (as *Eugenia myrtifolia*) cultures have been initiated from seed [111]. In vitro collections of *S. anisatum*, *S. australe*, *S. leuhmannii*, *S. moorei*, *S. paniculatum*, and *S. pseudofastigiatum*, all initiated from cutting material, are currently maintained at the Australian PlantBank (pp. 290–291, [71]). Initiation of cultures from seed, embryo, and embryonic axes of *S. fullagarii* and *S. unipunctatum* have also been trialed; however, sterilizing this material sufficiently for successful initiation has proven difficult (L. Hardstaff, unpubl.).

A few published reports of cryopreservation are available for this genus. Shatnawi et al. [110] reported successful cryopreservation of encapsulated shoot tips of *S. francissi* when the encapsulated material was precultured on 0.75 M sucrose media for one day followed by 6 h dehydration to 20% moisture content. Malik et al. [112] reported 100% survival following cryopreservation for embryonic axes of *S. cumini* cultured on media with 3% sucrose but details of the cryopreservation treatments used were not supplied. Evaluation of the use of droplet-vitrification, vacuum-infiltration vitrification, and encapsulation-dehydration to cryopreserve embryonic axes of *S. maire* found none of the methods resulted in ongoing embryo survival following exposure to liquid nitrogen [113]. *S. maire* embryonic axes have been shown to survive exposure to liquid nitrogen after encapsulation-dehydration; however, the embryos did not form complete plantlets after radicle elongation [108]. Trials of cryopreservation of *S. anisatum*, *S. australe*, *S. fullagarii*, and *S. paniculatum* embryonic axes or shoots have commenced at The Australian PlantBank and Kings Park and Botanic Garden, but no material has yet survived exposure to liquid nitrogen, regardless of pre-treatment (L. Hardstaff, E. Bunn, unpubl.).

### 5.3. Backhousia

The 13 known *Backhousia* species occur on the east coast of Australia, predominantly in Queensland [2]. *Backhousia citriodora* and *B. myrtifolia* are perhaps the most well-known species, the former for its value as a commercial species and the latter as a species commonly used in habitat restoration programs [77]. *Backhousia* spp. have been reported to have a range of susceptibility to myrtle rust, from ‘low’ to ‘very high’ [81]. Myrtle rust damage can impact conservation research on such species by reducing seed availability, as reported in a study of seed set, characteristics, longevity and germination of *B. citriodora* (intermediate) and *B. myrtifolia* (orthodox) [77]. No reports of successful cryopreservation could be found for this genus. Attempts to conserve the genus in vitro have been limited by difficulties with initiation, including limited availability of clean material and issues with contamination [114]. Previous studies have reported the difficulty of growing *B. citriodora* from cuttings, with root formation strongly linked to genotype [115], which may limit the conservation of genetic diversity within the species.

### 5.4. Gossia

*Gossia* is a recently described genus containing 39 species occurring in New Guinea, the Southwest Pacific and eastern Australia [2]. A number of species in this genus have been reported to have some degree of susceptibility to myrtle rust [81]. While the seeds of one species (the endangered *G. fragrantissima*) have been reported to be orthodox [78], recent surveys have found population decline and limited seed production in *A. psidii* infected populations of *G. hillii*, *G. lewisensis*, *G. inophloia*, and *G. punctata* [70]. Even before the advent of myrtle rust, Shapcott [116] observed very few seedlings and no viable seed production for the extremely rare *G. gonoclada*. Seeds for only two species, *G. fragrantissima* and *G. hillii*, are currently held in conservation seed banks [73]. No reports of in vitro culture or cryopreservation for the genus could be found in the literature; however, *G. fragrantissima* has been successfully propagated by cuttings and initiated into tissue culture at The Australian PlantBank [114].

### 5.5. Lenwebbia

*Lenwebbia* is another recently described genus, endemic to Australian rainforests, with only two described species and another two species yet to be published [117]. *Lenwebbia lasioclada* and *L. prominens* are found in north-eastern Australia [86]. *L. prominens* is thought to have orthodox seed storage behavior [78] and two seed collections are held at the Australian PlantBank. Both *L. lasioclada* and *L. prominens* are reported to have ‘high’ susceptibility to myrtle rust and the disease has been recorded on reproductive structures of *L. prominens* [81], which is likely to reduce the future availability of seed for research and conservation. *Lenwebbia* sp. Blackall Range and *L.* sp. Main Range are reported to have some variation in susceptibility but have low seedling recruitment [70]. Cutting-grown collections of the critically endangered *L.* sp. Main Range, which has exhibited severe population decline and very low reproductive capacity [70], have been established at the Australian PlantBank but the species has proven difficult to culture in vitro. While explants have survived initial cleaning, sterilizing and incubation on initiation medium, they have failed when transferred to multiplication media (WPM, MS or 1/2MS) [114].

### 5.6. Rhodamnia

*Rhodamnia* is a genus of 41 species found in China, Indo-China, Australia and the Southwest Pacific [2]. Of the 20 species found in north-eastern Australia, four are critically endangered under state and/or national legislation, including *R. angustifolia*, *R. longisepala*, *R. maideniana*, and *R. rubescens* [118]. Both *R. maideniana* and *R. rubescens* are listed as having ‘very high’ susceptibility to *A. psidii* [81] and were only recently listed as critically endangered after severe population declines caused by myrtle rust [53]. A study of *R. rubescens* in situ comparing control and fungicide-treated plots over a period of 24 months found that myrtle rust may reduce ongoing recruitment of the species, with infected flowers producing no fruit on the individuals studied. The authors of the study also observed a reduced abundance of seedlings under the canopy of control plots, with evidence of defoliation and mortality of seedlings caused by the disease [119]. Population studies of *R. spongiosa* and *R. pauciovulata* also found decline and very low reproductive capacity as a result of *A. psidii* [70]. While seed collections for four species (*R. argentea*, *R. dumicola*, *R. maideniana* and *R. rubescens*) are held in conservation seed banks [73], the seed of *R. maideniana* is known to be freezing-sensitive or short-lived in storage at −20 °C [78] and recent experiments on *R. rubescens* have also demonstrated short-lived behavior at that temperature (K. Sommerville, unpubl.). *R. rubescens* has been grown successfully in tissue culture and protocols for multiplying and deflasking the species have been developed [75], however there are no published reports of successful cryopreservation for this species.

### 5.7. Rhodomyrtus

Of the 21 *Rhodomyrtus* species distributed from tropical and sub-tropical Asia to the Southwest Pacific [2], 10 are found in north-eastern Australia. While other Australian species are not currently listed as threatened species, *R. psidioides* is listed as critically endangered after significant population decline caused by myrtle rust [53,120]. *R. psidioides* is listed as having ‘very high’ susceptibility to *A. psidii* and other Australian species have ‘moderate’ or ‘high’ susceptibility [81]. Repeated surveys of *R. psidioides* have found ongoing decline and even collapse of populations due to *A. psidii* infection causing death of mature trees and preventing seed production and seedling recruitment [59,70,119]. Given the urgent need for ex situ conservation and lack of fruit production in the wild, a seed orchard of *R. psidioides* was established at the Australian Botanic Garden Mount Annan [121]. Seed produced from the orchard was used to determine that the seeds were desiccation tolerant but freezing sensitive and did not have orthodox seed storage behavior, as previously thought [78]. *R. psidioides* has been grown successfully in vitro and protocols for multiplying and deflasking the species have been published (pp. 282–283, [71,75]). Cryopreservation trials have found the seeds to grow into healthy seedlings after osmotic desiccation and incubation in PVS2 for 30 min, but not surviving exposure to liquid nitrogen using a droplet vitrification protocol (L. Hardstaff, unpubl.). No reports of ex situ conservation could be found in the literature for other native Australian species, however successful in vitro induction of callus has been reported for *R. tomentosa* [122], which grows as an exotic species in Australia.

## 6. Discussion

The Myrtaceae is a large family with many species of economic, ecological and cultural importance. Increasing threats to this family from land clearing, cataclysmic wildfires, and myrtle rust are increasing the need for human intervention to avert the extinction of individual species [53]. While many dryland species in this family can be conserved by seed banking, the number of species falling in the category of ‘exceptional’ is growing, as research into seed storage behavior identifies species with recalcitrant, freezing-sensitive or short-lived seeds [75,76,77,78], and as myrtle rust reduces the capacity for seed set [59,70,123]. Long-term conservation of these exceptional species will require investment in cryobiotechnologies–in vitro culture, cryopreservation and supporting sciences.

The application of cryobiotechnologies to Myrtaceae genera native to subtropical and tropical rainforests–such as *Backhousia*, *Gossia*, *Lenwebbia*, *Rhodamnia*, *Rhodomyrtus* and *Syzygium*–may be challenging given that species in those genera are not likely to be adapted to either desiccation or chilling stresses. Cryopreservation of embryonic axes rather than shoot tips has been recommended for woody tropical species with desiccation sensitive seeds [124] and this strategy has also been used to preserve plumules from seeds of the temperate but desiccation-sensitive *Quercus robur* (Fagaceae) [125]. This technique may be appropriate for the large seeds of *Syzygium* species, but for genera with smaller, desiccation tolerant seeds (e.g., *Gossia* and *Rhodamnia*), cryopreservation of whole seeds may be more appropriate if seeds can be obtained. This strategy has been used successfully to preserve seeds of rainforest-origin plants such as Australian *Citrus* spp. (Rutaceae) [126], *Carica papaya* cultivars (Caricaceae) [127], and *Coffea* spp. (Rubiaceae) [128]. It is also appropriate for threatened Myrtaceae species with orthodox seeds. Sixteen Myrtaceae taxa have been cryopreserved using orthodox seed at Kings Park and Botanic Garden [98,129] and a further 60 species have recently been cryopreserved in the same manner at The Australian PlantBank [130]. Where seed is unavailable, however, it will be necessary to first initiate the species into in vitro culture, and develop suitable protocols for multiplication and deflasking, to provide shoot tips for cryopreservation.

Over 100 species and hybrids of Australian Myrtaceae are mentioned in the available literature as having been trialed in tissue culture. In contrast, only a handful of species have been reported as successfully cryopreserved using shoot tips [95,110,129] or embryonic axes [112]. Likewise, there are relatively few reports of successful cryopreservation using shoot tips of woody tropical species, but survival of 70% or more following cryopreservation has been reported for *Parkia speciosa*, *Trichilia emetica*, *Carica papaya*, *Citrus aurantium*, *Citrus suavissima* and *Manihot esculenta* [124]. Shoot tips from in vitro cultures have been also been used to cryopreserve the fungal-blight affected *Castanea dentata* (Fagaceae) [131]. Application of antioxidants such as vitamins C and E during pretreatment and recovery may help to increase survival and regrowth after cryopreservation of both shoot tips [132] and seed embryonic axes [133].

While some work has been done to preserve economically important species in the Myrtaceae family, there has been very little work utilizing cryobiotechnology to conserve threatened taxa. Given the devastating impacts of myrtle rust on a number of taxa in the wild, this is a field of research demanding urgent attention. The successful in vitro culture of threatened species in the genera *Gossia*, *Rhodamnia*, *Rhodomyrtus* and *Syzygium* provides a starting point for this research. Reported successes in cryopreservation of shoot tips of *S. francissi* [110] and embryonic axes of *S. cumini* [112], indicate that long-term conservation of exceptional Myrtaceae species is possible with the right protocol.

The development of successful cryopreservation protocols can be very complex, requiring an investigation of multiple factors affecting survival during pre-conditioning, freezing, and thawing processes [134]; the key challenge being to reduce cell water content sufficiently to prevent ice crystal formation without severely reducing cell viability. The development of suitable protocols for threatened Myrtaceae will require investigation of the effect of each stage of the cryopreservation process on biophysical and biochemical characteristics of the target germplasm. As recommended by Normah et al. [135], once tissue has been established in vitro, the effect on survival and regrowth of pretreatments, loading solutions, cryoprotectants, exposure to liquid nitrogen, rewarming, and unloading solutions, will need to be investigated sequentially. Characterizing changes in sugar content and cell membrane lipid composition [136,137], the activity of anti-oxidant enzymes [138], changes in oxidative status [139], and changes in metabolic rates [140] and mitochondrial function [141] in response to the application of pre-treatments and cryoprotectants may aid in identifying the most appropriate protocol for a given species. Tools such as differential scanning calorimetry [142,143] could be utilized to optimize tissue desiccation strategies such as vacuum infiltration vitrification, cryo-mesh or flash drying [144,145]. Ideally, these tools would be used to develop a protocol for each genus that could be successfully applied to multiple species in the genus with minimal variation.

Cryobiotechnology provides a viable alternative for the long-term conservation of exceptional species and a potential safe storage method for the many Myrtaceae species under threat from *A. psidii*. However, considerable gaps in knowledge will need to be addressed before cryopreservation techniques can be utilized for conservation on a routine basis. Application of available tools and collaboration between research groups working with similar species have the potential to expedite the development of successful storage protocols for species at imminent risk of extinction.

## Figures and Tables

**Table 1 plants-11-01017-t001:** Recorded variety of uses for Myrtaceae species in different industries.

Industry	Uses	Genus or Species	References
Plantation	Timber, pulp, fuel, charcoal	*Eucalyptus* spp. e.g., *E. camaldulensis*, *E. globulus*, *E. grandis* and *E. tereticornis* and their crosses	[1,10]
Agriculture	Windbreaks	*Eucalyptus*	[11]
	Pesticides	*Eucalyptus*, *Melaleuca*, and others (essential oils)	[10,12]
	Honey production	Various, mainly *Eucalyptus*	[1]
	Livestock breeding	*Syzygium aromaticum* (essential oils)	[13,14]
Horticulture	Ornamental species	*Syzygium*, *Callistemon*, and *Melaleuca*	[1]
	Cut flowers and foliage	*Chamelaucium* (flowers)	[15]
		*Eucalyptus* (foliage)	[16,17]
Medicine	Traditional medicines	*Eucalyptus pachyphylla* (flowers and sap)	[4]
		*Babingtonia camphorosmae* (flowers, leaves and stems), *Kunzea preissiana* (leaves and flowers), *Eucalyptus* and *Corymbia* (leaves and gum), *Melaleuca radula* (leaves)	[6]
		*Rhodomyrtus tomentosa* (flowers, fruit, leaves, bark, sap, roots)	[18]
		Various, including *Campomanesia*, *Eugenia* and *Myrcia*	[19]
	Diabetes	*Syzygium cumini* (extracts)	[20]
	Bacterial infections	*Corymbia torelliana*, *Melaleuca alternifolia* (extracts)	[21,22]
	Viral infections	*Melaleuca alternifolia*, *Backhousia citriodora* (extracts)	[23,24]
	Fungal infections	*Eucalyptus* (extracts)	[10]
	Mosquito control	Various, including *Eucalyptus* and *Melaleuca* (extracts)	[25,26,27,28]
Food	Fresh or processed fruit	*Eugenia* spp., *Kunzea pomifera*, *Myrciaria cauliflora*, *Psidium cattleyanum*, *P. guajava*, *Syzygium aqueum*, *S. cumini, S. jambos*, *S. leuhmannii*, *S. samarangense*	[1,29,30]
	Spices	*Backhousia citriodora* (leaves), *Pimenta dioica* (fruit), *Syzygium anisatum* (leaves), *S. aromaticum* (flower buds)	[1,29]
	Teas	*Backhousia citriodora*, *Melaleuca citrolens*, *Syzygium anisatum* (leaves)	[29,31]
	Additives (flavoring, antioxidant, antibacterial)	*Backhousia citriodora*, *Eucalyptus citriodora*, *E. olida*, *E. stragiana*, *Syzygium anisatum*, *S. leuhmannii* (extracts)	[10,29,32,33,34,35]

**Table 2 plants-11-01017-t002:** Reported use of cryobiotechnology for ex situ conservation of *Eucalyptus*.

Species	Propagule ^1^	Method ^2^	Success	References
*E. grandis*	Axillary bud	LN	Limited	[97]
*E. grandis × E. urophylla*	Callus	LN	Limited	[101,103]
*E. grandis*	Pollen	LN	Yes	[97]
*E. dunnii, E. urophylla* and *E. robusta*	Pollen	LN	Yes	[99]
*E. burracoppinensis, E. lane-poolei,* and *E. loxophleba* var. *gratiae*	Seed	LN	Limited	[98]
*E. microtheca*	Seed	LN	Yes	[100]
Various	Shoot tips	LN	Yes	[87]
Various	Seed and seedlings	SE	Yes	[88]
*E. globulus, E. saligna × E. maidenii*	TC material	SE	Yes	[88]
*E. grandis × E. urophylla*	Callus	IVC	Yes	[104]
*E. maculata*	Cutting	IVC	Yes	[105]
*E. dolorosa*, *E. graniticola*, *E. impensa* and *E.phylacis*	Cutting	IVC	Yes	[93,94,95]

^1^ TC material: shoot apices and leaves from first and second nodes of in vitro collections. ^2^ LN: cryopreservation in liquid nitrogen; SE: somatic embryogenesis; IVC: in vitro culture.

## Data Availability

Not applicable.

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
