# Peer review of "Myrtaceae in Australia: Use of Cryobiotechnologies for the Conservation of a Significant Plant Family under Threat"

_plants, 2022, doi:10.3390/plants11081017_

Round 1

Reviewer 1 Report

I would suggest a different more expanded title as half of this manuscript is a review of the Myrtaceae and is not about cryopreservation.

This is a very well written and interesting manuscript.  It has a very comprehensive review of the Myrtaceae genera and of rust on viability of many species (5 pages of text, too long). Section 3.1 could be reduced and included in the previous paragraph.  The section on cryopreservation is well done (4 pages of text). 

The only shortcoming of this manuscript is the discussion (1 page of text, too short). Interpretation of the existing research and discussing where to go next is the most important part of any review. The authors need to expand on, more specifically, the direction research needs to go. Providing the whole list of everything you could look at to improve recovery is not useful.  With the combined knowledge of the authors from their work with these plants, they should outline some more specific pathways that have not been explored (i.e. antioxidants), and the reasons that they are most likely to be the best ones to use in improving cryopreservation protocols.

Reviewer 2 Report

The manuscript presents a thorough appraisal of past and current efforts to develop cryotechniques aiming at Myrtaceae species preservation, with a particular focus placed on myrtle rust, identified as a rising concern, both in Australia and worldwide.

The manuscript is well documented and draws from ongoing research efforts yet unpublished and thus unavailable to the rest of the scientific community. The manuscript should definitely be published, although it might benefit from the following remarks.

The introduction is much too long; all the information presented is interesting and relevant but the core theme of the paper of the paper (i.e. cryopreservation) appears only on p6, which means much of the text should be reshaped and edited for concision.

The focus placed on myrtle rust-linked effects on Myrtaceae species (notably in the title) seems somewhat artificial as no specificity of this threat –compared to other risks- is clearly identified in the manuscript; in any case, those cryobiotechnologies to be involved are not specific to any given risk.

I would therefore just cite myrtle rust as an additional, though increasing, risk to natural populations in this family, unless the authors show that in some cases (to be clearly underlined) cryopreservation programmes were only implemented due to the disease.

Finally, I have mixed feelings about reporting research results from “personal communications”, especially in such numerous occurrences as in this manuscript. A significant number of these are rightly quoted in the text, but some appear in the literature list, which is definitely not standard procedure.

I agree that unpublished results can be of general/specific interest; obviously, although they certainly reflect ongoing research efforts accurately, they cannot be dealt with as published material is.

In the present manuscript, there are too many references to unpublished results/personal communications, so that the “literature review” feature is somewhat obscured; I wonder whether the manuscript should be written as a “state of the art” presentation, rather than a standard review, thus providing a coherent and comprehensive overview of ongoing projects on Myrtaceae biotechnologies (including cryotechniques) in Australia.

In conclusion, the manuscript is indeed of interest and should in my opinion be published. However, I would recommend shortening the text (especially the introduction) and presenting this work as a synopsis of current research efforts on Myrtaceae biotechnologies, especially those aimed at preserving species threatened with various hazards (including myrtle rust).

Reviewer 3 Report

The work entitled: “Cryopreservation of Australian Myrtaceae Species Susceptible to Myrtle Rust” might be of interest for both researchers working in the field of the Myrtaceae family, as well as cryobiologists.

However in order to be published it is in need of major revision.

There are some minor lexical and typological remarks to be addressed, such as:

Line 62: it has to be clear that “peppermint” stands for peppermint-scented, analogically to lemon-scented, above.

Lines 89 and 91 please insert the second bracket of the Latin names.

Abstract

Here, there should be shortly presented what motivated this review with the topic as it is, then a very short outline of the aspects covered by the review (significance of Myrtaceae in global and local aspects, threats to Myrtaceae, imposed by A. psidii, why we expect cryopreservation to assist the eradication of this pathogen, etc.). Then the conclusion stands well as it is now. The current content of the Abstract makes it to seem more as an Introduction, and then, Introduction itself needs more elaboration, as mentioned below.

Introduction

The main accent of the review, reflected in the title is cryopreservation approach of Myrtaceae sp. susceptible to myrtle rust. Then, logically, this needs to be reflected in the Introduction section in order to justify the motivation to perform this review. It has been reflected in the Abstract, but is needed to be supported by the respective references in the Introduction.

Line 52: the role of Myrtaceae species as firebrand tools for land management is not clear.

Review body-text

There is only one sub-chapter 3.1 of the chapter “3. Significance of Myrtaceae in Australia”and the rest of the information is presented as a general text without structuring.

Try to apply a tight and logic structuring of the whole content of the review. For example, here, in chapter 3, try to outline as sub-chapters all aspects of the significance of Myrtaceae in Australia which you have concerned and not only “3.1. The ‘Bush Food’ Industry”. The same applies to the rest of the chapters.

There are certain facts which need to be supported by references, such as: Line 117: “Threats to the Myrtaceae include changing land use, habitat fragmentation, increasing rates of natural disasters – such as extreme fires and flooding – due to climate change, and invasive species. Invasive pathogens, in particular pose a significant threat to the family”.

Chapter “6. Cryobiotechnology applied to Myrtacaeae”. Try to bring order to the content. Its structure is firstly set as: “1) two very large genera, i.e. Eucalyptus and Syzygium; 2) one small genus including species of commercial and ecological value, i.e. Backhousia [68]; and 3) four genera with a number of species at risk of imminent extinction from myrtle rust, i.e. Gossia, Lenwebbia, Rhodamnia, and Rhodomyrtus  [53,66,68].” Then the first sub-chapters are: “6.1 Backhousia”, then “6.2 Eucalyptus”, “6.3 Gossia”, and so on.

The potential of cryopreservation for eradicating A. psidii from regenerated plants has been concerned as a hypothesis in Chapter 6: “Cryobiotechnology could thus provide a viable alternative for the conservation of exceptional Myrtaceae species including the many species under threat from A. psidii.” Then concrete examples follow on the susceptibility of some of the species to A. psidii.

Then the work sets as a conclusion the hypothesis that “Cryobiotechnology provides a viable alternative for the long-term conservation of exceptional species and a potential safe storage method for the many Myrtaceae species under threat from A. psidii.”

However in order to meet the concept of the work this hypothesis needs to be supported of a detailed separately formulated chapter of the role of plant cryopreservation in pathogen eradication with concrete examples. In addition to this, if there is literature data on examples for A. psidii eradication by means of cryopreservation in any plant species this would significantly raise the merit of the work.

Round 2

Reviewer 3 Report

The authors have significantly improved the quality of the manuscripr, so now it can be published in the Plants Journal